# Learning Adaptive Lighting via Channel-Aware Guidance

**Qirui Yang** [*1 2] **Peng-Tao Jiang** [*2] **Hao Zhang** [2] **Jinwei Chen** [2] **Bo Li** [2] **Huanjing Yue** [1] **Jingyu Yang** [1]

## Abstract

Learning lighting adaptation is a crucial step in achieving good visual perception and supporting downstream vision tasks. Current research often addresses individual light-related challenges, such as high dynamic range imaging and exposure correction, in isolation. However, we identify shared fundamental properties across these tasks: i) different color channels have different light properties, and ii) the channel differences reflected in the spatial and frequency domains are different. Leveraging these insights, we introduce the channel-aware Learning Adaptive Lighting Network (LALNet), a multi-task framework designed to handle multiple light-related tasks efficiently. Specifically, LALNet incorporates color-separated features that highlight the unique light properties of each color channel, integrated with traditional color-mixed features by Light Guided Attention (LGA). The LGA utilizes color-separated features to guide color-mixed features focusing on channel differences and ensuring visual consistency across all channels. Additionally, LALNet employs dual domain channel modulation for generating color-separated features and a mixed channel modulation and light state space module for producing color-mixed features. Extensive experiments on four representative light-related tasks demonstrate that LALNet significantly outperforms state-of-the-art methods on benchmark tests and requires fewer computational resources. *We provide an online demo at LALNet.*

## 1. Introduction

Photography is the art of light. The quality of an image is crucial for effective visual presentation and robust performance in subsequent computer vision tasks. However,

---

[*]Equal contribution [1]Tianjin University, Tianjin, China. [2]vivo Mobile Communication Co., Ltd, Hangzhou, China. Correspondence to: Peng-Tao Jiang <pt.jiang@vivo.com>, Jingyu Yang <yjy@tju.edu.cn>.

*Proceedings of the 42nd International Conference on Machine Learning*, Vancouver, Canada. PMLR 267, 2025. Copyright 2025 by the author(s).

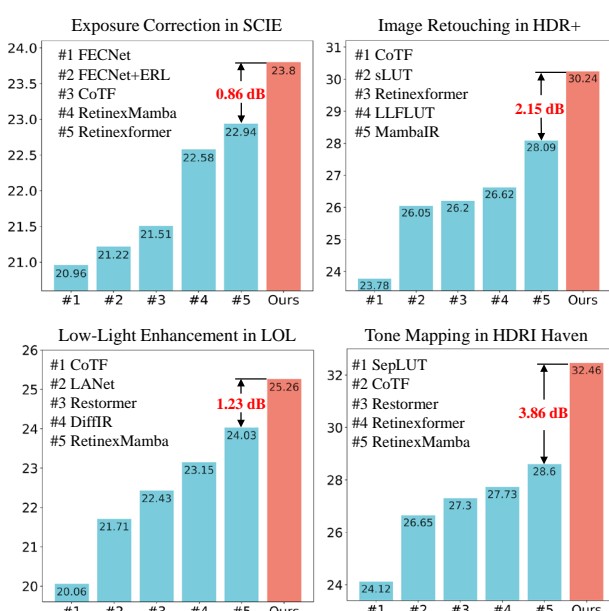

*Figure 1.* Our LALNet significantly outperforms state-of-the-art methods on four representative benchmark tests of light-related image enhancement, including image retouching, tone mapping, low-light enhancement, and exposure correction.

images taken under poor lighting conditions often exhibit degraded quality, which not only affects visual presentation but also poses challenges for tasks such as object detection and tracking. Consequently, learning adaptive lighting has emerged as a pivotal step in achieving robust visual perception and supporting downstream vision tasks. This process is analogous to the perception of the human visual system, that is, light adaptation, which enables us to maintain stable visual perception across diverse lighting environments.

Many tasks in computer vision aim to achieve light adaptation, including exposure correction (Li et al., 2024a; Huang et al., 2023), image retouching (He et al., 2020; Zhang et al., 2024), low-light enhancement (Cai et al., 2023; Yi et al., 2023), and tone mapping (Cao et al., 2023; Yang et al., 2022). The common goal of these light-related tasks is to adjust the light level of the scene to the perceptually optimal level, thereby revealing more visual details. However, due to the different characteristics of these light-related tasks, most of the current methods (Zeng et al., 2020; Li et al., 2024a) are designed to deal with the above tasks individually and are difficult to apply to other light-related tasks. For

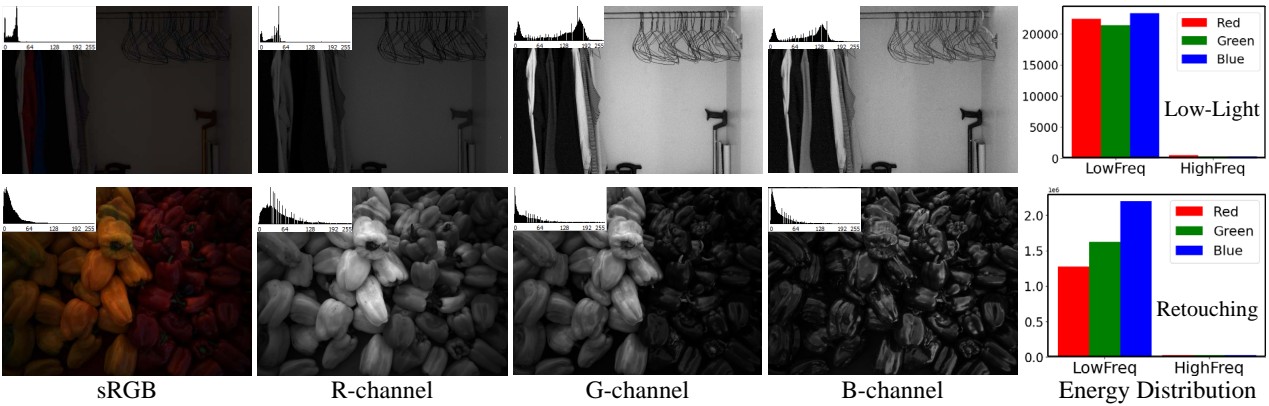

*Figure 2.* Motivation of LALNet. Different color channel differences and statistical DWT spectral energy distributions for different tasks.

example, exposure correction (Huang et al., 2022b; Zhang et al., 2019b) must adjust the brightness of both underexposed and overexposed scenes to achieve clearer images; image retouching (Wang et al., 2023; Su et al., 2024) aims to enhance the aesthetic visual quality of images affected by light defects, often requiring special attention to global light; low-light enhancement (Wang et al., 2022; Liu et al., 2021a) reveals more details by boosting the brightness of dark areas, but requires special processing of noise; and tone mapping (Zhang et al., 2022; Wang et al., 2021) preserves rich details by compressing high dynamic range light to low dynamic range, focusing more on adaptation to high dynamic range light. The different characteristics of these tasks make existing methods inconsistent in performance on multiple tasks. Although some works (Yang et al., 2023a) have attempted to perform light-related tasks with a unified architecture, the insufficient analysis of light-related task specificity has resulted in unsatisfactory performance compared to methods designed for these individual tasks.

*Interestingly, can a framework be designed to handle these light-related tasks, just as the human visual system can adapt to a variety of lighting environments?* Motivated by this question, we aim to design a framework capable of handling multiple light enhancement tasks separately.

To this end, we delve deep into analyzing the common light properties of these light-related tasks and utilize them to inspire the design of a multi-task framework. We observe two key insights from light-related tasks: **i) different color channels have different light properties**; **ii) the channel differences reflected in the spatial and frequency domains are different.** To analyze these differences, we employ the Discrete Wavelet Transform (Shensa et al., 1992) to decompose the input image into low-frequency and high-frequency components, and statistics on the energy distribution of the R/G/B channels based on the square of the pixel values separately. Fig. 2 illustrates the color channel attributes of two light-related task images in the spatial and frequency domains. It can be observed that the light proper-

ties of different channels differ significantly and that there is no fixed pattern between the different images. For example, for the first image, the G-channel exhibits a more balanced luminance distribution, while for the second image, the R-channel performs better in this regard. On the other hand, the frequency domain exhibits channel differences that are different from the spatial domain. For example, in the first image, the G-channel is brighter, but the G-channel does not have the highest energy distribution in the frequency domain. This illustrates that capturing channel differences in the spatial and frequency domains is different. Channel differences cannot be fully characterized in the spatial or frequency domains alone. Moreover, it is well known that the specific attributes (Yang et al., 2023a; Zhang et al., 2024) of light-related tasks are mainly embodied in the low-frequency components, whereas the details of the contents are more related to the high-frequency components. These findings highlight the importance of learning adaptive lighting by leveraging distinctive features of different color channels in the spatial and frequency domains.

Motivated by the above light properties, we propose a learnable adaptive lighting network, namely LALNet. Our method leverages the potential channel light differences to guide effective adaptive lighting. We decompose the light adaptation problem into two sub-tasks: (i) light adaptation, which addresses light variations under different light conditions, and (ii) detail enhancement, which preserves and refines image details while performing adaptive lighting. LALNet begins to learn adaptive light enhancement from the down-sampled version of the input image, optimizing for low computational complexity. To implement light adaptation, we propose a dual-branch architecture comprising channel separation and channel mixing. The channel separation branch employs the Dual Domain Channel Modulation module to extract color-separated features, focusing on light differences and color-specific luminance distributions for each channel in the spatial and frequency domains. In the channel mixing branch, we apply Mixed Channel Modulation and Light State Space Module to integrate color-mixed

lighting information, capturing inter-channel relationships and lighting patterns that achieve harmonious light enhancement. A key component of our framework is Light Guided Attention (LGA), which utilizes color-separated features to guide color-mixed light information for adaptive lighting. This mechanism enhances the network's capability to perceive changes in channel luminance differences and ensure visual consistency and color balance across channels. Consequently, our network is effectively adaptive to light variations while attending to feature differences across channels. Finally, we employ an iterative detail enhancement strategy to recover the image resolution level by level while enhancing the details. We conduct comprehensive experiments and demonstrate the state-of-the-art performance of our LALNet on four light-related tasks, as shown in Fig. 1. Our contributions can be summarized as follows:

- We propose a multi-task light adaptation framework inspired by the common light property, namely the Learning Adaptive Lighting Network (LALNet).

- We introduce the Dual Domain Channel Modulation to capture the light differences of different color channels and combine them with the traditional color-mixed features with Light Guided Attention.

- Extensive experiments on four representative light-related tasks show that LALNet significantly outperforms state-of-the-art methods in benchmarking and that LALNet requires fewer computational resources.

## 2. Related Work

**Exposure Correction.** Exposure correction aims to balance image brightness under varying lighting conditions (Yang et al., 2020; Nsampi et al., 2021; Huang et al., 2022b; Li et al., 2024b). Early methods like RetinexNet (Liu et al., 2021a) follow the Retinex theory to separately process illumination and reflectance. ZeroDCE (Guo et al., 2020) estimates pixel-wise curves without reference images. However, these approaches mainly address underexposure and struggle with diverse real-world scenarios. LPNet (Afifi et al., 2021) and FourierNet (Huang et al., 2022b) introduce multi-scale and frequency-aware designs for broader exposure handling. Recently, COTF (Li et al., 2024b) proposed a collaborative framework for real-time correction, effectively integrating global and pixel-level adjustments.

**Image Retouching.** Image retouching focuses on restoring natural luminance and color distributions in a more perceptually faithful manner (Moran et al., 2020; Liang et al., 2021a; Gao & Wu, 2021). Some approaches reformulate the problem as curve estimation (Kim et al., 2020; Li et al., 2020), while others like DeepLPF (Moran et al., 2020) optimize spatially adaptive filters for fine control. Lookup table (LUT)-based models (Zeng et al., 2020; Liang et al., 2021a)

offer efficient inference by learning compact representations. CSRNet (He et al., 2020) leverages conditional MLPs for adaptive enhancement, and GAN-based methods (Chen et al., 2018; Ni et al., 2020) enable unpaired learning, though often at the cost of interpretability and training stability.

**Tone Mapping.** Recent advancements in tone mapping have leveraged deep learning methods (Zhang et al., 2022; Yang et al., 2022; Zhang et al., 2024; Hu et al., 2022) to address the nonlinear mapping from HDR to LDR images. Hou et al. (Hou et al., 2017) applied CNNs to tone mapping tasks, establishing a foundation for subsequent research. Later works explored GANs for pixel-level accuracy (Cao et al., 2020; Rana et al., 2020; Panetta et al., 2021). Despite these advancements, issues such as halo artifacts and local inconsistencies persist. JointTM (Hu et al., 2022) combined tone mapping and denoising using discrete cosine transforms, while HSVNet (Zhang et al., 2019a) leveraged HSV color space manipulation to reduce halos and enhance detail retention. Despite notable progress, existing methods often struggle to balance global and local tone mapping, resulting in unsatisfactory results in other tasks.

**Low-Light Image Enhancement.** Recent advances in low-light enhancement are largely driven by deep learning (Yang et al., 2023b; Liu et al., 2021b; Yang et al., 2025). Models like DeepUPE (Wang et al., 2019b) and Retinexformer (Cai et al., 2023) build on Retinex theory for illumination decomposition. Hybrid architectures such as SNRNet (Xu et al., 2022) and Restormer (Zamir et al., 2022) incorporate Transformer designs for long-range dependencies. RetinexMamba (Bai et al., 2024) introduces the State Space Model to improve efficiency. However, Retinex-based methods (Cai et al., 2023; Liu et al., 2021a; Bai et al., 2024) are based on the theory of separated illumination and reflection, but they usually assume smooth and uniform lighting conditions, which may not hold in realistic scenes involving complex lighting variations. Moreover, these methods typically work in luminance or reflection space, where high-frequency details may be distorted during decomposition.

## 3. Methods

### 3.1. Motivation

Previous studies (Cai et al., 2023; Li et al., 2024a; Zhang et al., 2024; Su et al., 2024) for light-related tasks, such as tone mapping and low-light enhancement, are often tailored to individual tasks, leading to suboptimal performance across multiple scenarios. These frameworks typically fail to account for the common properties shared across different lighting-related tasks, which limits their generalizability. As a result, many frameworks are either overly specialized or inefficient when faced with multiple tasks. This leads to performance inconsistencies, especially when frameworks designed for specific tasks are applied to others. For in-

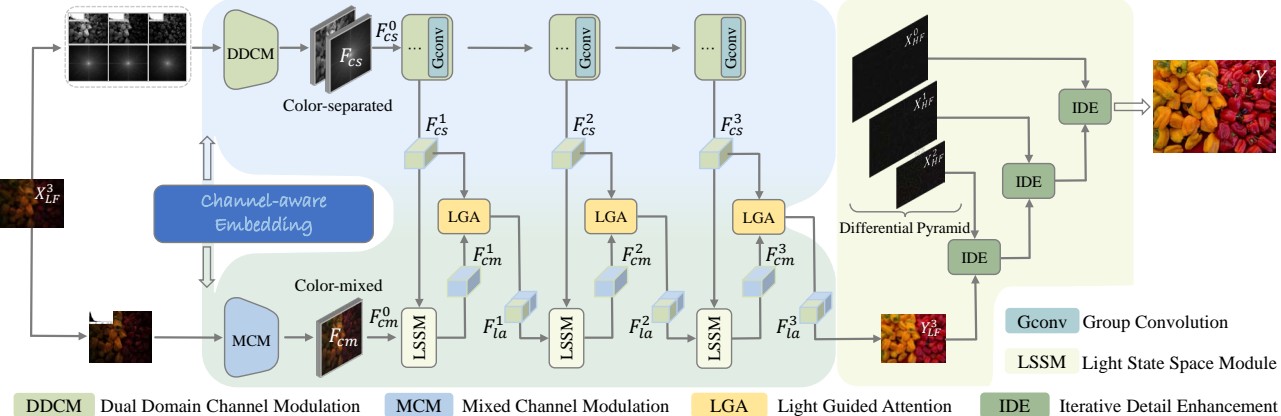

Figure 3. Architecture of LALNet for light adaptation. The core modules of LALNet are: (a) dual domain channel modulation (DDCM) that extracts color-separated features, focusing on light differences for each channel in the spatial and frequency domains, and (b) light guided attention (LGA) utilizes color-separated features to guide color-mixed light information for light adaptation.

stance, Retinexformer focuses on separating reflection and illumination to enhance low-light images, but its underlying Retinex theory is inapplicable to tasks such as tone mapping and image retouching. This limitation is evident in scenarios where low-light enhancement methods struggle to maintain color fidelity during tone mapping.

Our motivation is rooted in the observation that, despite the diverse nature of light-related tasks, there are key shared properties: **distinct light properties across color channels** and **channel differences in spatial and frequency domains.** These channel differences manifest differently in both the spatial and frequency domains, further complicating the task of adaptive lighting. To address these issues, we aim to design a multi-task framework that adapts to different lighting conditions more effectively than previous frameworks that focus on individual tasks. By analyzing these shared light properties across multiple tasks, our framework seeks to capture the subtle differences between color channels and ensure consistent and balanced visual outcomes across various lighting conditions.

### 3.2. Framework Overview

The overall pipeline of LALNet is illustrated in Fig. 3. LALNet is composed of two components: light adaptation and detail enhancement. Given a low-quality (LQ) input image $\mathbf{X}^0$, our goal is to generate a high-quality (HQ) output $\mathbf{Y}$ with optimal light. We begin to learn adaptive light from the down-sampled version of the input image $\mathbf{X}_{LF}^3$, optimizing for low computational complexity. Subsequently, we employ the two-branch structure for extracting light features, containing color separation and color mixing branches. The channel separation branch employs the DDCM and group convolution to extract color-separated feature $\mathbf{F}_{cs}$, focusing on light differences and color-specific luminance distributions for each channel in the spatial and frequency domains. In the channel mixing branch, we utilize mixed channel

modulation (MCM) combined with the light state space module (LSSM) to extract color-mixed feature $\mathbf{F}_{cm}$, promoting cross-channel interaction and achieving balanced light enhancement. This can be expressed mathematically as:

$$\mathbf{F}_{cs} = \mathrm{GConv}(\mathrm{DDCM}(\mathbf{X}_{LF}^3)), \tag{1}$$

$$\mathbf{F}_{cm} = \mathrm{LSSM}(\mathrm{MCM}(\mathbf{X}_{LF}^3), \mathbf{F}_{cs}). \tag{2}$$

To emphasize the light differences in different channels, we introduce Light Guided Attention, which injects the color-separated features into color-mixed features to obtain the light adaptive feature $F_{la}$, which is described as:

$$\mathbf{F}_{la} = \mathrm{LGA}(\mathbf{F}_{cm}, \mathbf{F}_{cs}). \tag{3}$$

This process ensures consistent and uniform light adaptation across the entire image and eliminates color distortion caused by channel crosstalk. Finally, we integrate the low- and high-frequency via learnable differential pyramid (Yang et al., 2024) and iterative detail enhancement, progressively refining image resolution and enhancing fine details.

### 3.3. Light Adaptation

In the literature, we generally utilize the traditional convolutions to convolve with all channels for light-related tasks, generating RGB-mixed features. This operator can capture the interaction information and shared features among channels. However, this also amplifies the luminance non-uniformity and noise existing in the three channels. Notably, for light-related tasks, we have observed that characteristic differences between the RGB channels and the spatial and frequency domains exhibit different differences. There is also no consistent pattern across images. As shown in Fig. 2, the three channels exhibit distinct differences in luminance, with one channel usually being closer to the ground truth. If we only utilize color-mixed features to adapt to light, the negative interference between channels will also

spread to all channels. Therefore, we introduce an additional branch that extracts channel-separated features alongside the channel-mixed features. Channel-mixed features are responsible for capturing mixed luminance and color information, while channel-separated features guide the network to focus on channel differences. This design prompts the network to adapt to light while attending to differences across channels.

### 3.3.1. COLOR SEPARATION REPRESENTATION

Based on the analysis in Sec. 1, the spatial and frequency domains reflect different channel differences. Therefore, we design DDCM to capture the color-separated features.

**Dual Domain Channel Modulation.** To avoid cross-channel interference between operating channels, we process each channel independently in the frequency and spatial domains and introduce learnable parameters to modulate the channels. After frequency domain processing, the images are inverted back to the spatial domain. Then, to complement the color-separated feature representation, we utilize channel attention to capture the color-separated features in the spatial domain. Specifically, given an input image $\mathbf{X}$, each channel of the image is denoted as $\mathbf{X}_i$ ($i = 1, 2, 3$). We perform a 2D fast Fourier Transform (FFT) for $\mathbf{X}_i$ to obtain the frequency domain representation:

$$\mathbf{S}_i(u, v) = \mathcal{F}(\mathbf{X}_i)(u, v) = \text{FFT2}(\mathbf{X}_i), \qquad (4)$$

where $\mathbf{S}_i(u, v) = \mathbf{R}_i(u, v) + j \cdot \mathbf{I}_i(u, v)$, $\mathbf{R}_i(u, v)$ and $\mathbf{I}_i(u, v)$ denote the real and imaginary parts, respectively. Then, we perform convolution operations on the $\mathbf{R}_i(u, v)$ and $\mathbf{I}_i(u, v)$, respectively:

$$\hat{\mathbf{R}}_i(u, v) = \mathbf{W}_{R_i} * \mathbf{R}_i(u, v), \quad \hat{\mathbf{I}}_i(u, v) = \mathbf{W}_{I_i} * \mathbf{I}_i(u, v), \qquad (5)$$

where $\mathbf{W}_{R_i}$ and $\mathbf{W}_{I_i}$ are the convolution kernels, $*$ denote convolution operation. Subsequently, we reorganize the decoupled real and imaginary parts into frequency-domain signals, and perform the Inverse Fourier Transform to obtain the decoupled time-domain information as follows:

$$\mathbf{S}'_i(u, v) = \mathbf{R}'_i(u, v) + j \cdot \mathbf{I}'_i(u, v), \qquad (6)$$

$$\mathbf{X}'_i = \mathcal{F}^{-1}(\mathbf{S}'_i(u, v)) = \text{IFFT2}(\mathbf{S}'_i). \qquad (7)$$

Finally, after concatenating channels, we capture the separated features of the image in the spatial domain through the channel attention module to further enhance the color-separated feature representation.

$$\mathbf{F}_{\text{cs}} = \text{CAB}(\text{Concat}(\mathbf{X}'_1, \mathbf{X}'_2, \mathbf{X}'_3)). \qquad (8)$$

### 3.3.2. COLOR MIXING REPRESENTATION

In parallel, we introduce mixed channel modulation for extracting channel-mixed features. Since light patterns often exhibit global characteristics (Rieke & Rudd, 2009; Yang et al., 2023a), inspired by (Finder et al., 2024), we employ wavelet transform to achieve channel-mixed features $\text{F}_{\text{cm}}$. The process begins with the extraction of small-scale features using a small convolutional kernel to capture local information. These features are then passed through a wavelet transform (WT), where the generated large-scale features modulate the small-scale features, enabling the network to better integrate global light representation. The process can be represented as follows:

$$\mathbf{cA}, \mathbf{cH}, \mathbf{cV}, \mathbf{cD} = \text{WT}(\text{Conv}_{3\times3}(\mathbf{X})), \qquad (9)$$

where $\mathbf{cA}, \mathbf{cH}, \mathbf{cV}, \mathbf{cD}$ represent the components of the 2D wavelet transform. Afterward, the modulated features are concatenated and further passed the convolutional layer.

$$\mathbf{F}^0_{\text{cm}} = \text{Conv}_{3\times3}(\text{Concat}(\mathbf{cA}, \mathbf{cH}, \mathbf{cV}, \mathbf{cD})). \qquad (10)$$

To enhance the network's capability to capture global light information, we introduce the Light State Space Module (LSSM), which supplements mixed-channel modulation. The LSSM is designed to efficiently capture long-range dependencies with lower computational overhead than transformer-based methods. For feature integration and expansion, LSSM begins by integrating channel-mixed features with channel-separated features. This integrated feature is then expanded to a dimensionality of $2C$ via a linear layer. Following this expansion, the feature is divided into two distinct components, $\mathbf{F}_1$ and $\mathbf{F}_2$, according to the channel dimensions. Therefore, the channel-separated feature $\mathbf{F}^1_{\text{cs}}$, along with the newly formed $\mathbf{F}_1$ and $\mathbf{F}_2$, serve as inputs to three parallel processing streams. First Stream: Feature $\mathbf{F}_1$ undergoes an initial expansion to $\eta C$ channels through a linear layer, followed by depth-wise convolution, SiLU, 2D selective scanning (SS2D) (Guo et al., 2024), and LayerNorm. This sequence refines the representation of $\mathbf{F}_1$, emphasizing its spatial and channel-wise characteristics. Second Stream: Feature $\mathbf{F}^1_{\text{cs}}$ is processed directly using SS2D, capturing comprehensive global context without additional transformations. Third Stream: Feature $\mathbf{F}_2$ is subjected only to SiLU, preserving its original characteristics while enabling non-linear transformations that enrich its representation. Subsequently, the global information extracted from the first two streams is fused. This fused information is then multiplied with the output of the third stream. By doing so, the LSSM effectively integrates detailed local and global light patterns, enhancing the overall sensitivity of the network to varying lighting conditions. The whole process can be represented as follows:

$$\mathbf{F}_1, \mathbf{F}_2 = \text{Chunk}(\text{Linear}(\mathbf{F}^0_{\text{cm}} + \mathbf{F}^1_{\text{cs}})), \qquad (11)$$

$$\mathbf{F}'_1 = \text{SS2D}(\text{SiLU}(\text{DWConv}(\mathbf{F}_1))) + \text{SS2D}(\mathbf{F}^1_{\text{cs}}) \qquad (12)$$

$$\mathbf{F}'_2 = \text{SiLU}(\mathbf{F}_2), \quad \mathbf{F}^1_{\text{cm}} = \text{MLP}(\text{LN}((\mathbf{F}'_1 \otimes \mathbf{F}'_2))), \qquad (13)$$

where $\text{Linear}(\cdot)$ denote linear projection, $\otimes$ denotes the Hadamard product.

### 3.3.3. LIGHT GUIDED ATTENTION

Although LSSM performs well in capturing long-range dependencies, it still faces problems such as local information forgetting and channel redundancy. Moreover, color mixed features ignore the feature differences between different channels, treating them equally in the network. However, in light-related tasks, we have observed significant differences between color channels, with no consistent pattern across images. These differences are crucial for adaptive lighting. For this reason, we propose to inject color-separated features into color-mixed features by light guided attention to perceive channel differences.

Specifically, for the first LGA module, we input the channel-mixed features $\mathbf{F}_{\text{cm}}^1$ from LSSM and the channel-separated features $\mathbf{F}_{\text{cs}}^1$ from group convolution into the LGA. Subsequently, the input $\mathbf{F}_{\text{cm}}^1$ is processed through a $1 \times 1$ convolution followed by a depthwise convolution, producing $\mathbf{K}$ and $\mathbf{V}$ tensors with doubled the number of channels. This can be expressed mathematically as:

$$\mathbf{K}, \mathbf{V} = \text{Conv}_{3\times3}(\text{Conv}_{1\times1}(\mathbf{F}_{\text{cm}}^1)). \quad (14)$$

The query $\mathbf{Q}$ is then generated from the channel-separated features $\mathbf{F}_{\text{cs}}^1$:

$$\mathbf{Q} = \text{Conv}_{3\times3}(\text{Conv}_{1\times1}(\text{GConv}_{3\times3}(\mathbf{F}_{\text{cs}}^1))). \quad (15)$$

We compute the attention weights by the dot product between $\mathbf{Q}$ and $\mathbf{K}$, normalized by the softmax function, and multiplied by $\mathbf{V}$ to obtain the updated features:

$$\text{Attention}(\mathbf{Q}, \mathbf{K}, \mathbf{V}) = \text{softmax}(\frac{\mathbf{Q}\mathbf{K}^T}{\sqrt{d_K}} \times \tau)\mathbf{V}, \quad (16)$$

where $d_K$ is the dimension of $\mathbf{K}$ and $\tau$ denotes the scaling factor. It can be remarked that we utilize channel-separated features as $\mathbf{Q}$ vectors to motivate the model to focus on channel differences. In summary, the design of LGA enhances the adaptive representation of image features in both spatial and channel dimensions and improves the network's ability to capture dependencies between image channels. After LGA processing, we can obtain the low-resolution light-adaption output $\mathbf{Y}_{\text{LF}}^3$. Subsequently, we utilize the iterative detail enhancement strategy to enhance the detail of $\mathbf{Y}_{\text{LF}}^3$, which is introduced in the following.

### 3.4. Detail Enhancement

To achieve faithful reconstruction, we apply a learnable differential pyramid (LDP) (Yang et al., 2024) to capture high-frequency details. Through LDP, we obtain the multi-scale high-frequency features $\mathbb{X}_{\text{HF}} = \{\mathbf{X}_{\text{HF}}^0, \ldots, \mathbf{X}_{\text{HF}}^{L-1}\}$, tapering resolutions from $H \times W$ to $\frac{H}{2^{L-1}} \times \frac{W}{2^{L-1}}$. $L$ denotes the number of pyramid levels ($L$=3 in our framework). Using the high-frequency information $\mathbb{X}_{\text{HF}}$ captured, we employ an iterative detail enhancement to progressively refine

the light-adaption image $\mathbf{Y}_{\text{LF}}^L$. Specifically, for the $l_{th}$ pyramid, we first up-sample the low-frequency image $\mathbf{Y}_{\text{LF}}^l$ and concatenate it with the HF component $\mathbf{X}_{\text{HF}}^{l-1}$, then feed it into a residual network to predict a refinement mask $\mathbf{M}^{l-1}$. This mask allows pixel-by-pixel refinement of the HF component, which is subsequently added to the up-sampling $\mathbf{Y}_{\text{LF}}^l$ to generate the reconstructed result of the current layer $\mathbf{Y}_{\text{LF}}^{l-1}$. The process at the $l_{th}$ pyramid is formulated as:

$$\mathbf{M}^{l-1} = \text{Res}(\text{Concat}(\text{Up}(\mathbf{Y}_{\text{LF}}^l), \mathbf{X}_{\text{HF}}^{l-1})), \quad (17)$$

$$\mathbf{Y}_{\text{LF}}^{l-1} = \text{Up}(\mathbf{Y}_{\text{LF}}^l) + (\mathbf{X}_{\text{HF}}^{l-1}\mathbf{M}^{l-1}), \quad (18)$$

where $\text{Res}(\cdot)$ and $\text{Up}(\cdot)$ denote the residual block and up-sampling, respectively.

## 4. Experiments

### 4.1. Experimental settings

**Datasets.** We evaluate our method on four representative light-related tasks: exposure correction (SCIE (Cai et al., 2018)), image retouching (HDR+ Burst Photography (Hasinoff et al., 2016)), low-light enhancement (LOL dataset (Wei et al., 2018)), and tone mapping (HDRI Haven (Yang et al., 2024). Following the settings of (Huang et al., 2022a) for SICE, it contains 1000 training images and 24 test images. The HDR+ dataset is a staple for image retouching, especially in mobile photography. We utilize 675 image sets for training and 248 for testing. The LOL dataset (Wei et al., 2018) contains 500 image pairs in total, with 485 pairs used for training and 15 test images. The HDRI Haven dataset is a new benchmark for evaluating tone mapping (Su et al., 2021; Cao et al., 2023), which includes 570 HDR images of diverse scenes under various light conditions. We select 456 image sets for training and 114 for testing.

**Implementation details.** We implement our model with Pytorch on the NVIDIA L40s GPU platform. The model is trained with the Adam optimizer ($\beta_1 = 0.9$, $\beta_2 = 0.999$) for $4 \times 10^5$ iterations. The learning rate is initially set to $1 \times 10^{-4}$. We adopt traditional PSNR and SSIM metrics on the RGB channel to evaluate the reconstruction accuracy. We also employ TMQI (Yeganeh & Wang, 2013), LPIPS (Zhang et al., 2018), and CIELAB color space (Zhang et al., 1996) to evaluate image quality and perceptual quality, respectively.

### 4.2. Comparison with State-of-the-Arts

**Quantitative comparison.** The performance of the proposed multi-task framework is evaluated on four light-related image enhancement tasks, namely, (1) exposure correction, (2) image retouching, (3) low-light enhancement, and (4) tone mapping. We quantitatively compare the proposed method with a wide range of state-of-the-art light-related methods in Tab. 1, Tab. 2, Tab. 3, and Tab. 4. For exposure correction, as shown in Tab. 1, our method

*Table 1.* Quantitative results of exposure correction methods on the SCIE dataset. "/" denotes the unavailable source code. Metrics with ↑ and ↓ denote higher better and lower better. The best and second results are in red and blue, respectively.

| Method | Exposure Correction in SCIE | | | | | | | | |
| | Under | | Over | | Average | | | | |
| | PSNR↑ | SSIM↑ | PSNR↑ | SSIM↑ | PSNR↑ | SSIM↑ | LPIPS↓ | NIQE↓ | MUSIQ↑ |
|---|---|---|---|---|---|---|---|---|---|
| URtinexNet (Wu et al., 2022) | 17.39 | 0.6448 | 7.40 | 0.4543 | 12.40 | 0.5496 | 0.3549 | 12.78 | 49.11 |
| DRBN (Yang et al., 2020) | 17.96 | 0.6767 | 17.33 | 0.6828 | 17.65 | 0.6798 | 0.3891 | 12.06 | 48.77 |
| SID (Chen et al., 2018) | 19.51 | 0.6635 | 16.79 | 0.6444 | 18.15 | 0.6540 | 0.2417 | 11.79 | 51.07 |
| CSRNet (He et al., 2020) | 21.43 | 0.6789 | 20.13 | 0.7250 | 20.78 | 0.7019 | 0.1390 | 10.59 | 61.79 |
| MSEC (Afifi et al., 2021) | 19.62 | 0.6512 | 17.59 | 0.6560 | 18.58 | 0.6536 | 0.2814 | / | / |
| SID-ENC (Huang et al., 2022a) | 21.30 | 0.6645 | 19.63 | 0.6941 | 20.47 | 0.6793 | 0.2797 | 11.49 | 52.29 |
| DRBN-ENC (Huang et al., 2022a) | 21.89 | 0.7071 | 19.09 | 0.7229 | 20.49 | 0.7150 | 0.2318 | 11.23 | 54.15 |
| CLIP-LIT (Liang et al., 2023) | 15.13 | 0.5847 | 7.52 | 0.4383 | 11.33 | 0.5115 | 0.3560 | / | / |
| FECNet (Huang et al., 2022b) | 22.01 | 0.6737 | 19.91 | 0.6961 | 20.96 | 0.6849 | 0.2656 | 11.05 | 53.73 |
| FECNet+ERL (Huang et al., 2023) | 22.35 | 0.6671 | 20.10 | 0.6891 | 21.22 | 0.6781 | / | / | / |
| Retinexformer (Cai et al., 2023) | 23.75 | 0.7157 | 22.13 | 0.7466 | 22.94 | 0.7310 | 0.1714 | 10.37 | 55.67 |
| CoTF (Li et al., 2024a) | 22.90 | 0.7029 | 20.13 | 0.7274 | 21.51 | 0.7151 | 0.1924 | 10.19 | 51.61 |
| RetinexMamba (Bai et al., 2024) | 23.56 | 0.7212 | 21.59 | 0.7384 | 22.58 | 0.7298 | 0.1856 | 10.35 | 53.67 |
| LALNet-Tiny | 23.86 | 0.7197 | 22.26 | **0.7510** | 23.06 | 0.7354 | **0.1280** | **8.93** | **63.01** |
| LALNet | **24.63** | **0.7270** | **22.95** | 0.7473 | **23.80** | **0.7372** | 0.1397 | 9.34 | 61.49 |

*Table 2.* Quantitative results of image retouching methods. "/" denotes the unavailable source code.

| Method | #Params | Image Retouching in HDRPlus | | | | | | |
| | | PSNR↑ | SSIM↑ | TMQI↑ | LPIPS↓ | △E↓ | NIQE↓ | MUSIQ↑ |
|---|---|---|---|---|---|---|---|---|
| HDRNet (Gharbi et al., 2017) | 482K | 24.15 | 0.845 | 0.877 | 0.110 | 7.15 | 10.47 | 68.73 |
| CSRNet (He et al., 2020) | 37K | 23.72 | 0.864 | 0.884 | 0.104 | 6.67 | 10.99 | 67.82 |
| DeepLPF (Moran et al., 2020) | 1.72M | 25.73 | 0.902 | 0.877 | 0.073 | 6.05 | 10.35 | 70.02 |
| LUT (Zeng et al., 2020) | 592K | 23.29 | 0.855 | 0.882 | 0.117 | 7.16 | 11.36 | 67.67 |
| CLUT (Zhang et al., 2022) | 952K | 26.05 | 0.892 | 0.886 | 0.088 | 5.57 | 11.19 | 67.39 |
| sLUT (Wang et al., 2021) | 4.52M | 26.13 | 0.901 | / | 0.069 | 5.34 | / | / |
| SepLUT (Yang et al., 2022) | 120K | 22.71 | 0.833 | 0.879 | 0.093 | 8.62 | 12.26 | 67.89 |
| Restormer (Zamir et al., 2022) | 26.1M | 25.93 | 0.900 | 0.883 | 0.050 | 6.59 | 10.49 | 68.92 |
| LLFLUT (Zhang et al., 2024) | 731K | 26.62 | 0.907 | / | 0.063 | 5.31 | / | / |
| CoTF (Li et al., 2024a) | 310K | 23.78 | 0.882 | 0.876 | 0.072 | 7.76 | 11.54 | 68.07 |
| Retinexformer (Cai et al., 2023) | 1.61M | 26.20 | 0.910 | 0.879 | 0.046 | 6.14 | 10.75 | 68.93 |
| RetinexMamba (Bai et al., 2024) | 4.59M | 26.81 | 0.911 | 0.880 | 0.047 | 5.89 | 10.52 | 69.02 |
| MambaIR (Guo et al., 2024) | 4.31M | 28.09 | 0.943 | 0.879 | 0.028 | 5.31 | 10.76 | 70.05 |
| LALNet-Tiny | 230K | 29.30 | 0.939 | 0.886 | 0.030 | 5.04 | 9.70 | 69.98 |
| LALNet | 2.45M | **30.24** | **0.944** | **0.888** | **0.027** | **4.52** | 9.82 | **70.25** |

improves **2.29 dB** PSNR and 0.0221 SSIM compared to the CoTF (Li et al., 2024a) (CVPR24) method. For image retouching, as shown in Tab. 2, the proposed LALNet outperforms all the previous SOTA methods by a large margin. Specifically, our method significantly outperforms the SOTA methods MambaIR (Guo et al., 2024), RetinexFormer (Cai et al., 2023), LLFLUT (Zhang et al., 2024) and CoTF (Li et al., 2024a), RetinexMamba (Bai et al., 2024), improving PSNR by **2.15 dB** in the HDR+ dataset. LALNet-Tiny is a lightweight variant of LALNet (fewer feature channels and fewer LSSM blocks.) LALNet-Tiny has only $230K$ parameters and 1.75 GFLOPs, but the performance is also significantly better than other SOTA methods. For low-light enhancement, our LALNet significantly outperforms SOTA methods on the LOL-v1 dataset while requiring moderate computational and memory costs. Compared to the best recent method RetinexMamba, LALNet improves PSNR by **1.23 dB** and SSIM by 0.027, and LALNet only costs **16%** (6.70 / 42.82) GFLOPs. For tone mapping, Tab. 3 reports

*Table 3.* Quantitative results of tone mapping methods. "/" denotes the unavailable source code.

| Method | Tone Mapping in HDRI Haven | | | | |
| | PSNR↑ | SSIM↑ | TMQI↑ | LPIPS↓ | △E↓ |
|---|---|---|---|---|---|
| UPE (Wang et al., 2019a) | 23.58 | 0.821 | 0.917 | 0.191 | 10.85 |
| HDRNet (Gharbi et al., 2017) | 25.33 | 0.912 | 0.941 | 0.113 | 7.03 |
| CSRNet (He et al., 2020) | 25.78 | 0.872 | 0.928 | 0.153 | 6.09 |
| DeepLPF (Moran et al., 2020) | 24.86 | 0.939 | 0.948 | 0.077 | 7.64 |
| LUT (Zeng et al., 2020) | 24.52 | 0.846 | 0.912 | 0.171 | 7.33 |
| CLUT (Zhang et al., 2022) | 24.29 | 0.836 | 0.908 | 0.169 | 7.08 |
| LPTN (Liang et al., 2021b) | 26.21 | 0.941 | 0.954 | 0.113 | 8.82 |
| SepLUT (Yang et al., 2022) | 24.12 | 0.854 | 0.915 | 0.165 | 8.03 |
| Restormer (Zamir et al., 2022) | 27.30 | 0.954 | 0.948 | 0.032 | 5.67 |
| CoTF (Li et al., 2024a) | 26.65 | 0.935 | 0.948 | 0.098 | 5.84 |
| Retinexformer (Cai et al., 2023) | 27.73 | 0.955 | 0.949 | 0.030 | 5.41 |
| RetinexMamba (Bai et al., 2024) | 28.60 | 0.955 | 0.953 | 0.032 | 5.12 |
| LALNet-Tiny | 31.17 | 0.962 | 0.959 | 0.026 | 4.23 |
| LALNet | **32.46** | **0.969** | **0.961** | **0.019** | **3.58** |

the quantitative results on the HDRI Haven dataset. We can see that our method has the best overall performance.

**Qualitative results.** Visual comparison of LALNet and state-of-the-art light-related image enhancement methods are shown in Fig. 4, Fig. 5, Fig. 9, and Fig 10. Please zoom in for better visualization. To better visualize the

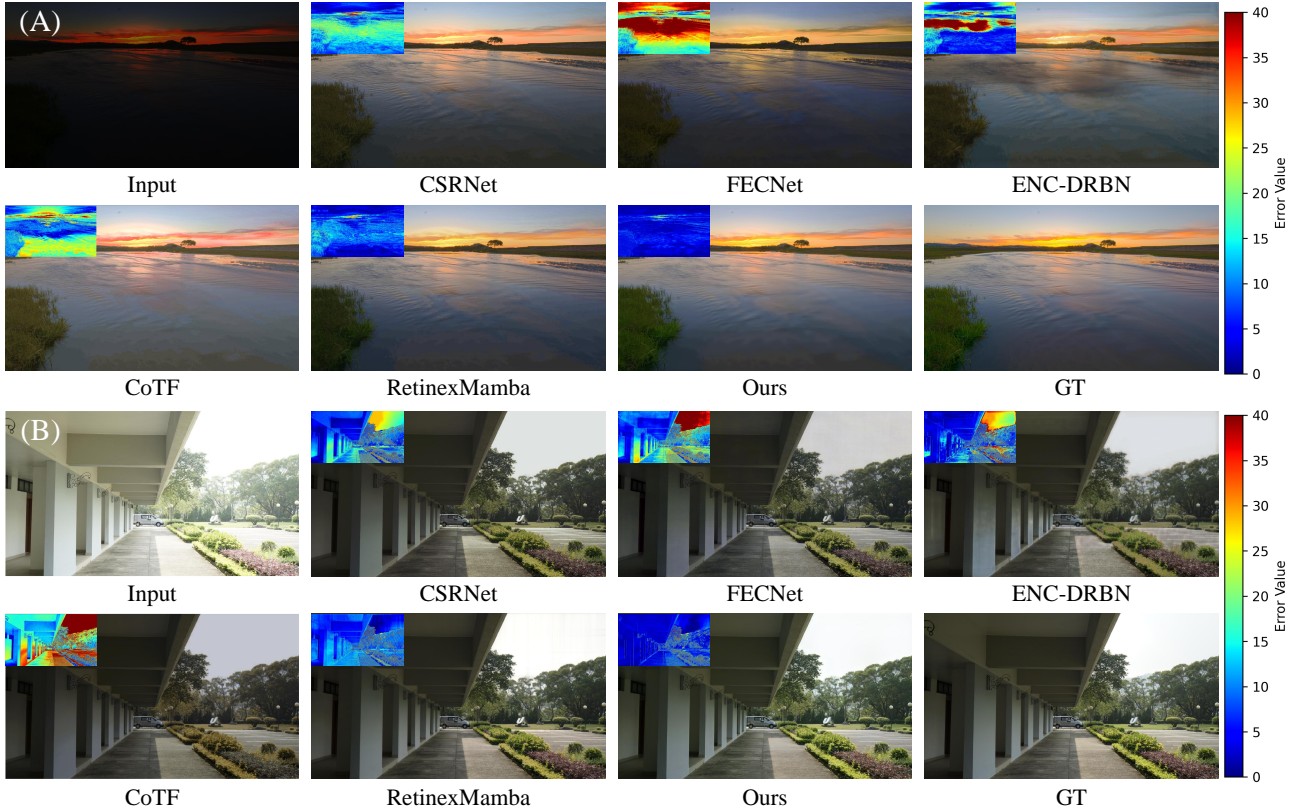

*Figure 4.* Visual comparisons between our LALet and the SOTA methods on the SCIE dataset. (Zoom in for best view.) The error maps in the upper left corner facilitate a more precise determination of performance differences.

performance differences of various methods, we present an error map to show the differences between the results of each method and the target image, as shown in the upper left corner of the image. In the error map, the red area indicates a larger difference, while the blue area indicates that the two are closer. Notably, error maps have no special units and only indicate errors. These figures illustrate that our LALNet consistently delivers visually appealing results on light-related tasks. Results reveal that the proposed method usually obtains better precise color reconstruction and vivid color saturation. Meanwhile, our method faithfully reconstructs fine high-frequency textures. For instance, in Fig. 4, the newest method, CoTF, exhibits distortion and color cast, but our LALNet still performs well. In Fig. 9, our method exhibits excellent color fidelity and restores proper global brightness and local contrast, consistent colors, and sharp details. These results prove that our method produces more pleasing visual effects. More results and visual comparisons are presented in our Appendix and LALNet.

## 4.3. Ablation studies

We conduct comprehensive breakdown ablations to evaluate the effects of our proposed framework.

**Effectiveness of specific modules.** To validate the effec-

*Table 4.* Quantitative results of LLE methods on the LOLv1 dataset. "*" denotes that the results are from reference papers.

| Method | GFLOPs | Low-Light Enhancement | |
|---|---|---|---|
| | | PSNR↑ | SSIM↑ |
| DeepUPE (Wang et al., 2019b) | 21.10 | 14.38 | 0.446 |
| DeepLPF (Moran et al., 2020) | 5.86 | 15.28 | 0.473 |
| UFormer (Wang et al., 2022) | 12.00 | 16.36 | 0.771 |
| RentinexNet (Wei et al., 2018) | 587.47 | 17.19 | 0.589 |
| EnGAN (Jiang et al., 2021) | 61.01 | 17.48 | 0.650 |
| Sparse (Yang et al., 2021) | 53.26 | 17.20 | 0.640 |
| FIDE (Xu et al., 2020) | 28.51 | 18.27 | 0.665 |
| KinD (Zhang et al., 2019b) | 34.99 | 20.35 | 0.813 |
| MIRNet (Zamir et al., 2020) | 785 | 24.14 | 0.842 |
| LANet (Yang et al., 2023a) | / | 21.71 | 0.810 |
| Restormer (Zamir et al., 2022) | 144.25 | 22.43 | 0.823 |
| CoTF (Li et al., 2024a) | 1.81 | 20.06 | 0.755 |
| Retinexformer (Cai et al., 2023)* | 15.57 | 23.93 | 0.831 |
| Diff-Retinex (Yi et al., 2023) | 396.32 | 21.98 | 0.852 |
| DiffIR (Xia et al., 2023) | 51.63 | 23.15 | 0.828 |
| RetinexMamba (Bai et al., 2024) | 42.82 | 24.03 | 0.827 |
| LALNet-Tiny | **1.75** | 24.07 | 0.845 |
| LALNet | **6.70** | **25.26** | **0.855** |

tiveness of the MCM, DDCM, LGA, and LSSM modules, we set up different variants to validate the effectiveness of the proposed framework. The results are listed in Tab. 5. Variants #1 serve as the baseline model and represent the removal of all modules and replacement with residual blocks. For Variants #2 apply a convolution block to replace the

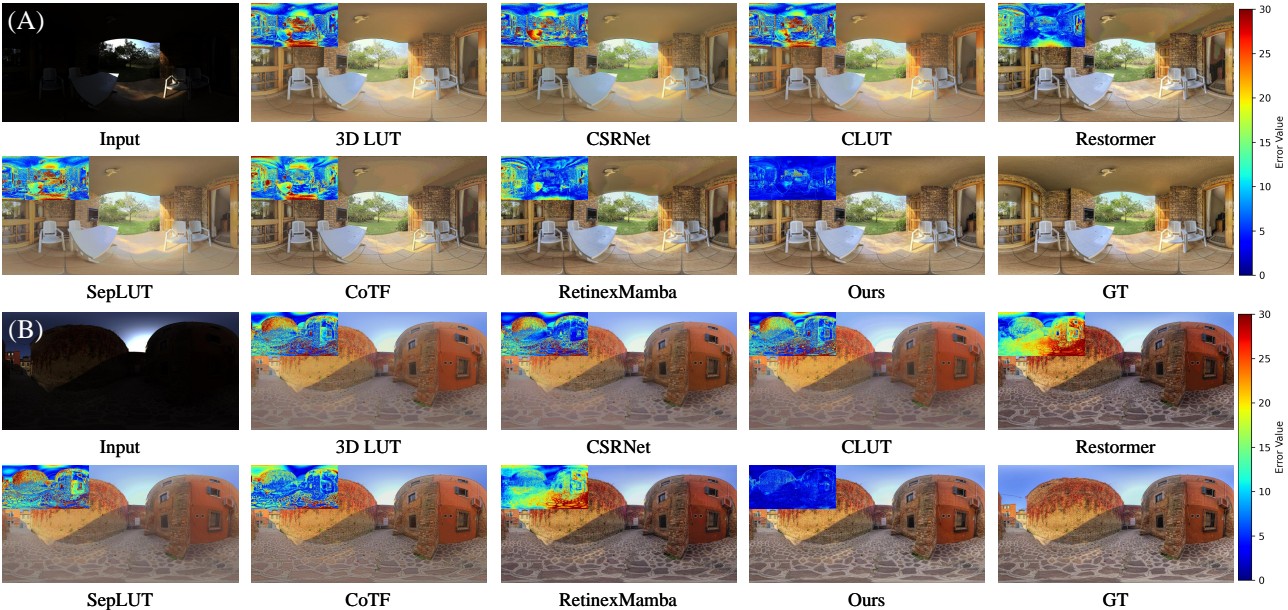

*Figure 5.* Visual comparisons between our LALet and the state-of-the-art methods on the HDRI Haven dataset (Zoom-in for best view). The error maps in the upper left corner facilitate a more precise determination of performance differences.

*Table 5.* Ablation studies of key components on SCIE dataset.

| Variants | MCM | DDCM | LGA | LSSM | PSNR↑ | SSIM↑ |
|---|---|---|---|---|---|---|
| #1 | ✗ | ✗ | ✗ | ✗ | 20.29 | 0.6834 |
| #2 | ✗ | ✓ | ✓ | ✓ | 23.95 | 0.7137 |
| #3 | ✓ | ✗ | ✓ | ✓ | 23.47 | 0.7144 |
| #4 | ✓ | ✓ | ✗ | ✓ | 23.53 | 0.7229 |
| #5 | ✓ | ✓ | ✓ | ✗ | 22.81 | 0.7091 |
| #6 | ✓ | ✓ | ✓ | ✓ | 24.63 | 0.7270 |

*Table 6.* Ablation study on the pyramid levels number. The "N.A." result is not available due to insufficient GPU memory.

| Metrics | Number of Levels | | | |
|---|---|---|---|---|
| | n=1 | n=2 | n=3 | n=4 |
| PSNR | N.A. | 23.45 | **24.63** | 23.07 |
| SSIM | N.A. | 0.7102 | **0.7270** | 0.7094 |
| TMQI | N.A. | 0.8735 | **0.8667** | 0.8783 |
| LPIPS | N.A. | 0.1240 | **0.1270** | 0.116 |
| △E | N.A. | 8.18 | **7.53** | 8.22 |
| #Params | **2.20M** | 2.29M | 2.45M | 2.81M |
| FLOPs | 36.16G | 12.20G | 6.70G | **5.51G** |

MCM with a performance reduction of 0.65 dB PSNR. In Variants #3, we use group convolution replacing DDCM to extract channel-separated features, and the PSNR is reduced by 1.16 dB. For Variants #4, we remove the LGA module and directly sum channel-mixed and channel-separated features for light guidance. The results confirm the effectiveness of the color-separated feature to guide the light adaptation, with a PSNR increase of 1.11 dB. For Variants #5, we replace the LSSM module with residual blocks and the performance drops by 1.82 dB. The results show that our proposed DDCM, LSSM, LGA, and MCM are effective compared to conventional feature extraction. These results consistently demonstrate the effectiveness of our method.

**Selection of the number of levels.** We validate the influence of the number of pyramid levels $l$. As shown in Tab. 6, the model achieves the best performance on all tested resolutions when $l = 3$. When a larger number of levels ($l \geq 4$) results in a significant decline in performance. This is because when $l$ is larger and the number of downsamples is more, the model fails to reconstruct the high frequencies efficiently, resulting in performance degradation. When $l = 1$, the low-frequency image resolution equals the input

image resolution, leading to a burst of computational memory. Comparing $l = 2$ and $l = 3$ demonstrates that despite the small input image resolution of the low-frequency pathway, high-frequency details can still be recovered efficiently in our framework.

## 5. Conclusion

This paper proposes a unified framework for learning adaptive lighting via light property guidance. In particular, we propose DDCM for extracting color-separated features and capturing the light difference across channels. The LGA utilizes color-separated features to guide color-mixed features for adaptive lighting, achieving color consistency and color balance. Extensive experiments demonstrate that our method significantly outperforms state-of-the-art methods, improving PSNR by 0.86 dB in the SCIE dataset, 2.15 dB in the HDR+ dataset, 1.12 dB in the LOL dataset, and 3.86 dB in the HDRI Haven dataset respectively compared with the second-best method.

## Acknowledgement

This work is supported by the National Natural Science Foundation of China under Grants 62231018 and 62072331.

## Impact Statement

The proposed LALNet framework represents a significant advancement in the field of image processing, offering a versatile and efficient solution for various light-related tasks. By adapting the training data, LALNet makes it easier for even non-experts to process high-quality images, contributing positively to areas where image quality is crucial. In addition, this research can further inspire developments in areas such as autonomous driving, where complex problems are solved in novel ways.

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

# Appendix

In this Appendix, we provide additional results and analysis.

## A. Further Analysis of Motivation

Different wavelengths of light exhibit different response characteristics when an image sensor captures photons for photoelectric conversion. After processing by an image signal processor, these differential responses are sometimes amplified or minimized but are difficult to eliminate. In addition, the differences in the Bayer pattern of different image sensors also result in different channels showing different responses to luminance and noise. Meanwhile, light sources in natural scenes are usually non-uniform, which also leads to the fact that sunlight, shadows, reflections, and other factors can cause RGB channels to respond differently to the same scene.

Recall that in Sec. 1, we discussed two observations that serve as the motivation to design LALNet. We show more motivation cases in Fig. 6 (the exposure correction and tone mapping tasks). In particular, (a) different color channels have different light properties, and (b) the channel differences reflected in the time and frequency domains are different. To further analyze our first motivation, we visualized the frequency domain images of the different channels using the Fourier Transform and compared them. The results show that, as in the time domain, significant differences are exhibited between the different channels in the frequency domain. Based on the observations in Fig. 2 and Fig. 6, the common properties of several light-related tasks investigated in this paper are verified, which also contribute to the design of our network.

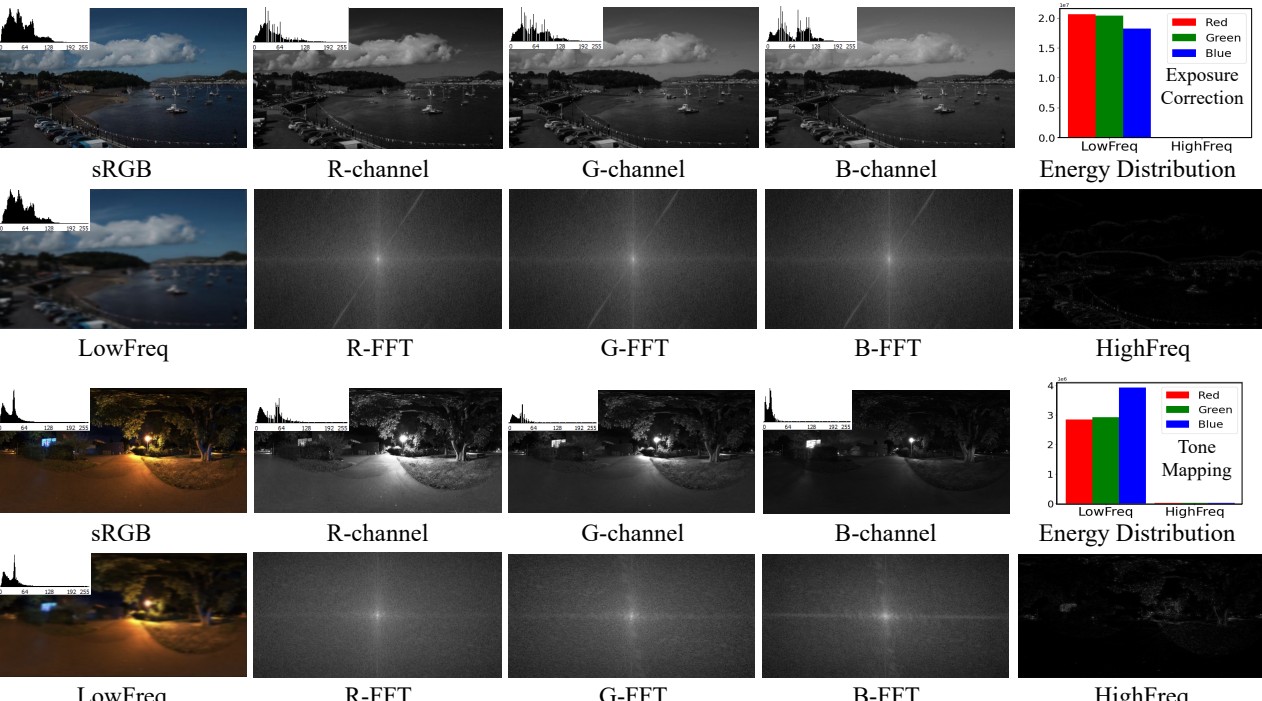

*Figure 6.* Motivation. Visualization of the light-related task images in different color channels and their corresponding DWT spectra energy distribution. R-FFT denotes the Fourier Frequency Domain diagram of the R channel. LowFreq and HighFreq are low-frequency and high-frequency images.

## B. Visualization in the Network

We demonstrate the Iterative Detail Enhancement modules (IDE), and Light State Space Module (LSSM) in Fig. 7 and Fig. 8. To reduce the computational resources, we implement light adaptation at low resolution. To compensate for the loss of details, we use an iterative detail enhancement module to recover high-frequency details. Specifically, as shown in Fig. 7, we first up-sample the low-frequency mapped image $\mathbf{Y}^i_{LF}$ and concatenate it with the HF component $\mathbf{X}^{i-1}_{HF}$, then feed it into a residual network to predict the mask $\mathbf{M}_{i-1}$. This mask allows pixel-by-pixel refinement of the HF component, which is subsequently added to the up-sampling $\mathbf{Y}^i_{LF}$ to generate the reconstructed result of the current layer $\mathbf{Y}^{i-1}_{LF}$.

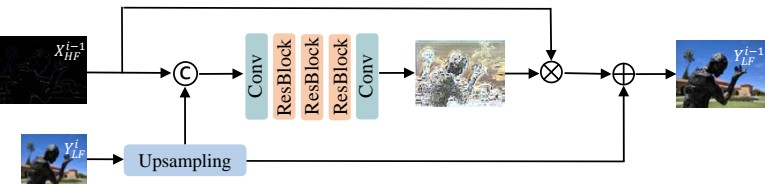

*Figure 7.* The architecture of the Iterative Detail Enhancement module progressively restores resolution and fine details.

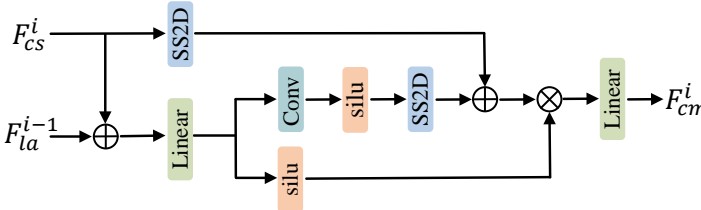

*Figure 8.* The architecture of the Light State Space Module.

## C. More Results

We further evaluate the effectiveness of our model on the exposure correction (Afifi et al., 2021), HDR Survey (Fairchild, 2023), and UVTM (Cao et al., 2023) datasets, all of which present more complex lighting conditions.

The MSEC dataset (Afifi et al., 2021) provides images rendered with relative exposure values (EVs) ranging from -1.5 to +1.5, comprising 17,675 training images, 750 validation images, and 5,905 test images. Table 7 presents the quantitative results on MSEC. As shown, our method achieves the best overall performance, with a PSNR of 23.93 dB, SSIM of 0.8734, and LPIPS of 0.0791.

*Table 7.* Quantitative results of exposure correction methods on the MSCE dataset.

| Method | Exposure Correction in MSCE | | | | | | |
| | Under | | Over | | Average | | |
| | PSNR↑ | SSIM↑ | PSNR↑ | SSIM↑ | PSNR↑ | SSIM↑ | LPIPS↓ |
|---|---|---|---|---|---|---|---|
| He (Pizer et al., 1987) | 16.52 | 0.6918 | 16.53 | 0.6991 | 16.53 | 0.6959 | 0.2920 |
| CLAHE (Reza, 2004) | 16.77 | 0.6211 | 14.45 | 0.5842 | 15.38 | 0.5990 | 0.4744 |
| WVM (Fu et al., 2016) | 18.67 | 0.7280 | 12.75 | 0.645 | 15.12 | 0.6780 | 0.2284 |
| RetinexNet (Wei et al., 2018) | 12.13 | 0.6209 | 10.47 | 0.5953 | 11.14 | 0.6048 | 0.3209 |
| URtinexNet (Wu et al., 2022) | 13.85 | 0.7371 | 9.81 | 0.6733 | 11.42 | 0.6988 | 0.2858 |
| DRBN (Yang et al., 2020) | 19.74 | 0.8290 | 19.37 | 0.8321 | 19.52 | 0.8309 | 0.2795 |
| SID (Chen et al., 2018) | 19.37 | 0.8103 | 18.83 | 0.8055 | 19.04 | 0.8074 | 0.1862 |
| MSEC (Afifi et al., 2021) | 20.52 | 0.8129 | 19.79 | 0.8156 | 20.08 | 0.8145 | 0.1721 |
| SID-ENC (Huang et al., 2022a) | 22.59 | 0.8423 | 22.36 | 0.8519 | 22.45 | 0.8481 | 0.1827 |
| DRBN-ENC (Huang et al., 2022a) | 22.72 | 0.8544 | 22.11 | 0.8521 | 22.35 | 0.8530 | 0.1724 |
| CLIP-LIT (Liang et al., 2023) | 17.79 | 0.7611 | 12.02 | 0.6894 | 14.32 | 0.7181 | 0.2506 |
| FECNet (Huang et al., 2022b) | 22.96 | 0.8598 | 23.22 | 0.8748 | 23.12 | 0.8688 | 0.1419 |
| LCDPNet (Zhang et al., 2019b) | 22.35 | 0.8650 | 22.17 | 0.8476 | 22.30 | 0.8552 | 0.1451 |
| FECNet+ERL (Zamir et al., 2020) | 23.10 | 0.8639 | 23.18 | 0.8759 | 23.15 | 0.8711 | / |
| CoTF (Yang et al., 2023a) | 23.36 | 0.8630 | 23.49 | 0.8793 | 23.44 | 0.8728 | 0.1232 |
| LALNet | **23.78** | **0.8638** | **24.01** | **0.8787** | **23.90** | **0.8713** | **0.0801** |

To further demonstrate the robustness and generalization ability of our model, we evaluate it on third-party non-homologous HDR image (HDR Survey) and video (UVTM) datasets, as summarized in Table 8. The HDR Survey dataset comprises 105 HDR images and is widely adopted for benchmarking HDR tone mapping methods (Cao et al., 2020; Rana et al., 2020; Panetta et al., 2021; Liang et al., 2018; Paris et al., 2011), though it does not provide ground-truth references. Likewise, the UVTM dataset contains 20 real-world HDR videos, also without ground truth. It is important to note that both the HDR Survey and UVTM datasets are used solely for testing purposes. As shown in Table 8, our method significantly outperforms existing approaches on both benchmarks in terms of TMQI. Specifically, our model achieves a TMQI score of 0.9296 on the HDR Survey and 0.9584 on the UVTM, surpassing all competing methods.

*Table 8.* Validating generalization on third-party datasets, including HDR Survey and UVTM video datasets.

| Datasets | Metrics | HDRNet | CSRNet | 3D LUT | CLUT | SepLUT | IVTMNet | CoTF | Ours |
|----------|---------|--------|--------|--------|------|--------|---------|------|------|
| HDR Survey | TMQI | 0.8641 | 0.8439 | 0.8165 | 0.8140 | 0.8085 | 0.9160 | 0.8612 | **0.9296** |
| UVTM | TMQI | 0.8281 | 0.8973 | 0.8787 | 0.8799 | 0.8629 | 0.8991 | 0.9006 | 0.9584 |

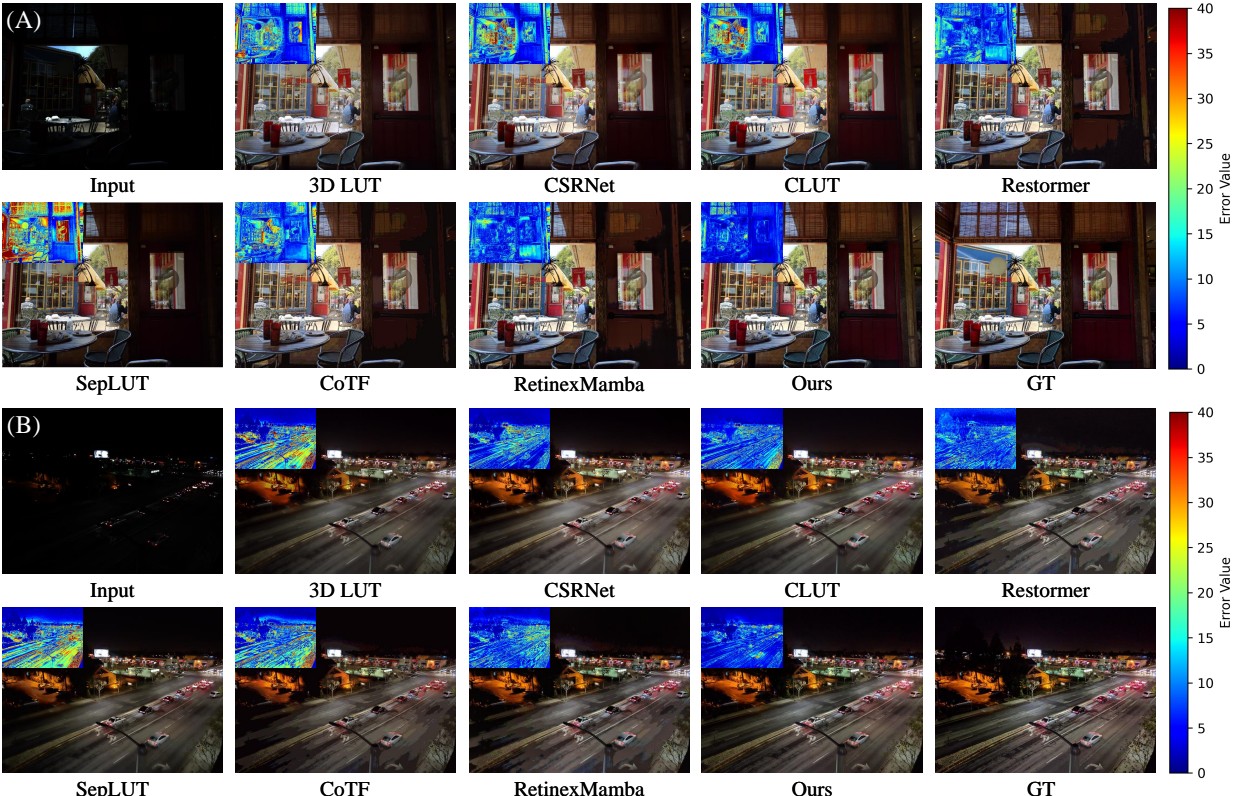

*Figure 9.* Visual comparisons between our LALet and the state-of-the-art methods on the HDR+ dataset.

# D. Ablation Study

To validate the effectiveness of the SS2D module, we use Self-Attention and Residual Block to replace the SS2D module in the original published model. We use the Self-Attention module released by Restormer (Zamir et al., 2022), and ResBlock is constructed from two convolutional layers and activation functions. The results, as shown in Table 9, show that using SS2D as part of the base module effectively captures global features and strikes a balance between performance and efficiency. Notably, the same excellent results are obtained using the Self-Attention module, which is attributed to the design of our overall framework, further demonstrating the effectiveness of our proposed adaptive lighting framework.

Further, we use DDCM to capture color-separated features, and to avoid channel mixing during information propagation, we use group convolution to keep the color channels separated. To verify the effectiveness of the design, we use traditional convolution to replace group convolution. The experimental results are shown in Table 10, where the channel mixing caused by the conventional convolution leads to a performance degradation. This phenomenon shows the necessity of color channel separation and the effectiveness of using color-separated features to guide light adaptation.

*Table 9.* Ablation study on the LSSM modules.

| Variants | Replaced Modules | #Params | FLOPs | PSNR↑ | SSIM↑ | TMQI↑ | LPIPS↓ | △E↓ |
|----------|------------------|---------|-------|-------|-------|-------|--------|-----|
| #1 | ResBlock | 2.99M | 7.13G | 22.81 | 0.7091 | 0.8635 | 0.1291 | 8.480 |
| #2 | Self-Attention | 2.25M | 6.48G | 24.41 | **0.7253** | 0.8657 | **0.1257** | **7.525** |
| #3 | Ours | 2.45M | 6.70G | **24.62** | 0.7227 | **0.8667** | 0.1297 | 7.529 |

*Table 10.* Ablation study on the group Convolution (G-Conv) and traditional Convolution (T-Conv).

| Variants | Replaced Modules | #Params | FLOPs | PSNR↑ | SSIM↑ | TMQI↑ | LPIPS↓ | △E↓ |
|----------|------------------|---------|-------|-------|-------|-------|--------|-----|
| #1 | T-Conv | 2.50M | 6.73G | 23.98 | 0.7121 | 0.8521 | 0.1363 | 8.146 |
| #2 | G-Conv | 2.45M | 6.70G | 24.62 | 0.7227 | 0.8667 | 0.1297 | 7.529 |

## E. Loss functions

The proposed framework obtains faithful light enhancement by optimizing the reconstruction loss, perceptual loss, and high-frequency loss. We utilize three objective losses to optimize our network, including reconstruction loss ($L_{\mathrm{Re}}$ and $L_{\mathrm{SSIM}}$), perceptual loss ($L_{\mathrm{P}}$), and high-frequency loss ($L_{\mathrm{HF}}$). To summarize, the complete objective of our proposed model is combined as follows:

$$L_{\mathrm{total}} = \alpha \cdot L_{\mathrm{Re}} + \beta \cdot L_{\mathrm{SSIM}} + \gamma \cdot L_{\mathrm{HF}} + \eta \cdot L_{\mathrm{P}}, \tag{19}$$

where $\alpha$, $\beta$, $\gamma$, and $\eta$ are the corresponding weight coefficients.

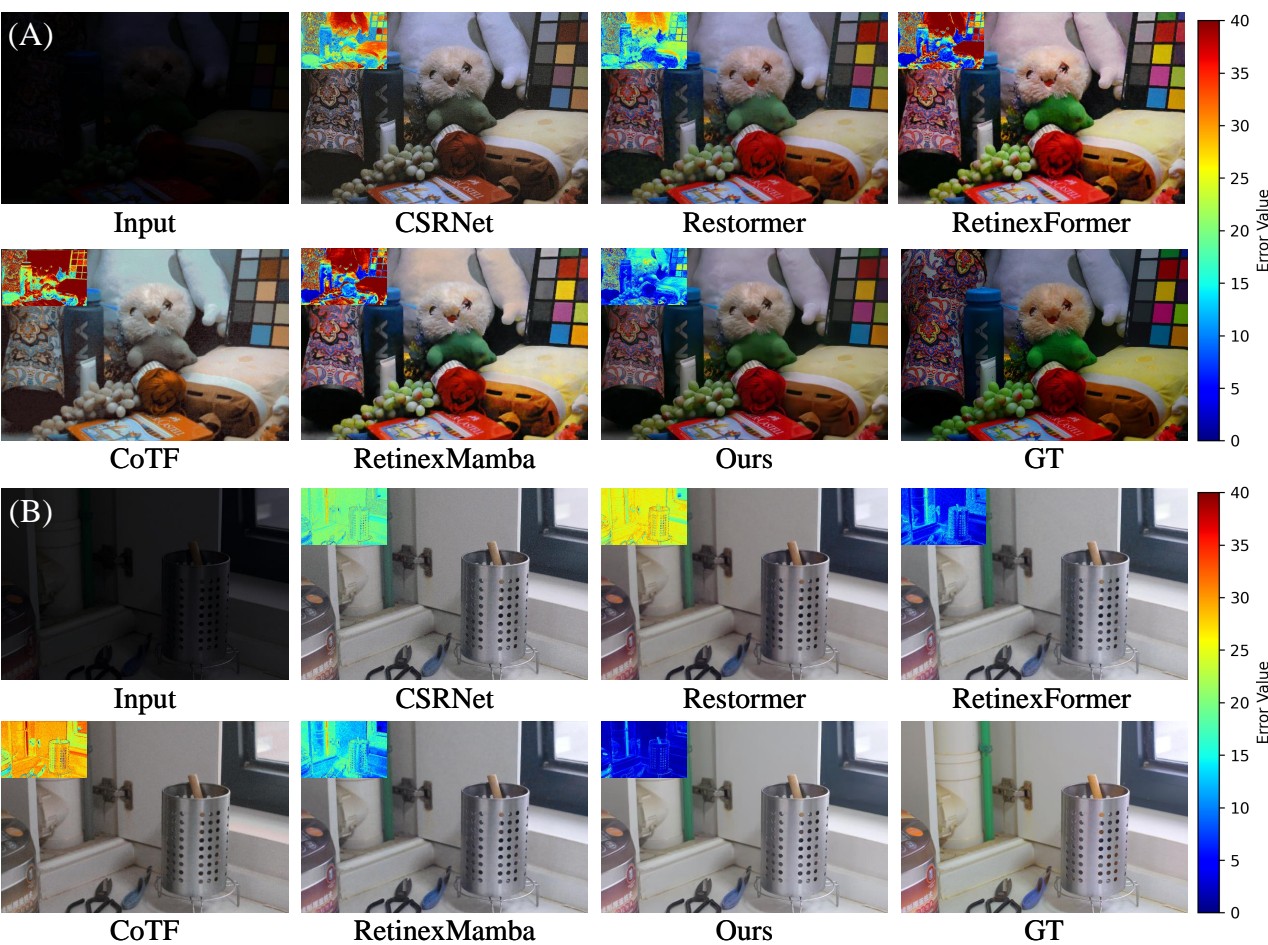

*Figure 10.* Visual comparisons between our LALet and the SOTA methods on the LOLv1 dataset.

