# OpenReview forum: "Learning Adaptive Lighting via Channel-Aware Guidance"
_ICML.cc/2025/Conference — ICML 2025 poster_

### Official Review · Reviewer_UEvw · 2025-03-09

**Overall Recommendation:** 4

**Summary:**

The paper uses the channel differences in spatial and frequency domains for lighting adaptation and mixes them for enhancement. The paper is well organised and the experimental results show the effectiveness and efficiency of the proposed method. The authors can give more detail about the structure of DDCM and MCM in Figure 3, which is not required in the rebuttal period.

**Claims And Evidence:**

Yes

**Essential References Not Discussed:**

No

**Experimental Designs Or Analyses:**

Yes

**Methods And Evaluation Criteria:**

Yes

**Other Comments Or Suggestions:**

No

**Other Strengths And Weaknesses:**

The paper is generally well-written.

**Questions For Authors:**

1. What do cA, cH, cV and cD mean in equation 11?
2. The structure of Color Mixing Representation and Light Guided Attention seems basic SSM(Mamba) and Attention structure, do you make any improvements?
3. How about the running speed of the proposed method for each image? Does it have advantages over other methods?
4. What are the differences between LALNet-Tiny and LALNet?

**Relation To Broader Scientific Literature:**

The paper can promote the development of image post-process.

**Theoretical Claims:**

Yes

---

> ### Author Rebuttal · Authors · 2025-03-29
>
> We sincerely appreciate the reviewer’s insightful comments and provide detailed responses below.
>
> >Q1. What do cA, cH, cV and cD mean in equation 11?
>
> **A1:** These variables represent the components of the 2D wavelet transform applied to the input image:
> 1. cA denotes the approximation coefficients of the wavelet transform.
> 2. cH denotes the horizontal detail coefficients.
> 3. cV denotes the vertical detail coefficients.
> 4. cD denotes the diagonal detail coefficients.
>
> The wavelet transform decomposes the image into different frequency bands, which enables effective separation of illumination and detail information. These components are then leveraged in our color mixing representation to enhance adaptive lighting adjustment. We will clarify this in the revised manuscript to improve readability.
>
> >Q2. The structure of Light Guided Attention seems basic SSM(Mamba) and Attention structure, do you make any improvements?
>
> **A2:** We acknowledge the reviewer’s concerns regarding the relationship between our Light State Space Module (LSSM) and Mamba. While LSSM shares similarities with state-space modeling principles, we introduce several key improvements beyond MambaIR:
> 1. Gated MLP instead of CAB: Unlike MambaIR, which uses the CAB module to mitigate channel redundancy, we found through activation visualizations that such redundancy was not significant in our task. Instead, we propose a Gated MLP module, which jointly processes spatial and channel-wise features, enhancing adaptability.
> 2. Channel separation in LSSM: We inject channel-separated features into the light adaptation process, enabling the network to better perceive channel brightness variations, thereby improving color consistency and balance across RGB channels.
> 3. LSSM as an optional component: While LSSM enhances feature extraction, it is not irreplaceable. We experimented with substituting LSSM with residual blocks and self-attention modules, and the model still achieved performance on par with state-of-the-art methods.
>
> Additionally, LGA utilizes channel-separated features to guide color-mixed features, ensuring effective lighting adaptation while maintaining cross-channel consistency—a novel application not explored in prior works. We will provide clearer explanations and appropriate citations in the revision.
>
>
> >Q3. How about the running speed of the proposed method for each image? Does it have advantages over other methods?
>
> **A3:** We recognize the importance of computational efficiency and provide a running time comparison with state-of-the-art methods.
>
> Table R3: Inference Speed Comparison (Seconds per Image on 480p).
> | Method          | Run Time |
> |:-:|:-:|
> | FECNet          | 0.012    |
> | COTF            | 0.010    |
> | CSRNet          | 0.012    |
> | RetinexMamba    | 0.4778   |
> | RetinexFormer   | 0.028    |
> | MambaIR         | 0.4595   |
> | Ours            | 0.018    |
>
> Our method achieves a competitive inference time of 0.018s per image, which is faster than RetinexFormer, RetinexMamba, and MambaIR, while maintaining superior enhancement quality. This demonstrates a favorable balance between efficiency and effectiveness.
>
> >Q4. What are the differences between LALNet-Tiny and LALNet?
>
> **A4:** LALNet-Tiny is a lightweight variant of LALNet, designed to improve efficiency while preserving strong performance. The key differences include:
> * Reduced channel dimensions: Fewer feature channels to reduce computational complexity.
> * Fewer LSSM modules: A more compact design with fewer LSSM modules.
>
> We will include a brief section describing LALNet-Tiny design and relationship to LALNet.
>
> We appreciate the reviewer’s recognition of our contributions and their thoughtful feedback. We believe these clarifications and improvements will further strengthen our work. Thank you for your time and valuable insights.

---

### Official Review · Reviewer_kmer · 2025-03-10

**Overall Recommendation:** 3

**Summary:**

This paper proposes a unified light adaptation framework, Learning Adaptive Lighting Network (LALNet), based on the finding that the color statistics on spatial and frequency domains differ from different light-related tasks. On top of this observation, the paper introduces a Dual Domain Channel Modulation module that captures and aggregates color-separated and color-mixed embeddings for optimal light to the target task. The method shows superior performance by evaluating the model on four light-related datasets, such as exposure correction, image retouching, low-light enhancement, and tone mapping.

**Claims And Evidence:**

The main finding of the paper is the color statistics on spatial and frequency domains differ from different light-related tasks. This observation leads to a unified framework that learns adaptive light for various light-related tasks. The finding and its corresponding architecture design sound good to me. And its effectiveness is well supported by experimental results.

**Essential References Not Discussed:**

The related works on a unified framework, low-light enhancement, exposure correction, and tone mapping are adequately cited in the main paper and supplementary material.

**Ethical Review Concerns:**

No ethical concerns

**Experimental Designs Or Analyses:**

The experiments are conducted on four different light-related tasks. One concern was that the used dataset seems small-scale, so the scalability of the proposed method is curious. But, supplementary material (Sec E) shows more complex and larger-scale experiments on low-light enhancement, HDR survey, and UVTM dataset. So, the overall experimental design and setup are sound good to me.
However, the LALNet-Tiny is not described in the manuscript how designed from LALNet.

**Methods And Evaluation Criteria:**

The paper proposed a unified light adaptation framework for four light-related tasks, including exposure correction, image retouching, low-light enhancement, and tone mapping. For this purpose, the paper evaluates the method on exposure correction (SCIE (Cai et al., 2018)), image retouching (HDR+ Burst Photography (Hasinoff et al., 2016)), low-light enhancement (LOL dataset (Wei et al., 2018)), and tone mapping (HDRI Haven (Zhang et al., 2019a), along with common image evaluation metrics, such as PSNR, SSIM, etc.
The proposed method, dataset, and evaluation criteria make sense for the paper's problem definitions.

**Other Comments Or Suggestions:**

* I think I might have missed it, But I can not find the loss function to train the model.
* LALNet-Tiny is not described in the manuscript how designed from LALNet.

**Other Strengths And Weaknesses:**

[Weakness]
* The presentation for the method section can be improved by adding subsection and subsubsection. For instance, the color mixing representation section is composed of the wavelet transform part, the Light State Space Module (LSSM) part, and its first, second, and third streams. They can be divided rather than describing them in a single paragraph. This is the same for other method sections. In general, the current presentation reduces the readability of the method sections.

**Questions For Authors:**

Please answer the weakness (Q1), other comments (Q2, Q3)

**Relation To Broader Scientific Literature:**

The paper's key contribution is a unified framework for various light-related tasks, which can influence low-light enhancement, exposure correction, HDR, retouching, etc.

**Theoretical Claims:**

No theoretical claims.

---

> ### Author Rebuttal · Authors · 2025-03-30
>
> We sincerely appreciate the reviewer’s constructive feedback and valuable suggestions. Below, we address the key concerns and outline our planned improvements.
>
> >Q1. Improving the Readability of the Method Section
>
> **A1:** To improve the clarity of the Method section, we have restructured it into a more logical and intuitive organization. The revised structure is as follows:
> * 2.1 Motivation: We introduce the fundamental challenges of lighting adaptation and explain why a unified framework is necessary.
> * 2.2 Framework Overview: We provide a high-level description of the proposed framework, detailing its architectural design and how different components interact.
> * 2.3 Color Separation Representation: Our model first extracts color-separated features to decouple illumination-related information in both the frequency and spatial domains. This step enables the network to learn distinct representations for different color channels, mitigating color distortion and channel-dependent illumination inconsistencies.
> * 2.4 Color Mixing Representation: We introduce mixed channel modulation for capturing inter-channel relationships and lighting patterns that achieve a harmonious light enhancement.
>    - 2.4.1 Mixed Channel Modulation: This subsection introduces mechanisms to effectively integrate cross-channel lighting information.
>   - 2.4.2 Light State Space Module (LSSM): We introduce the LSSM, which leverages state-space representations to refine the global illumination representation dynamically, improving stability and adaptability under varying lighting conditions.
> * 2.5 Light-Guided Attention: We propose a novel light-guided attention mechanism that utilizes color-separated features to guide color-mixed light information for adaptive lighting. This mechanism enhances the network’s capability to perceive changes in channel luminance differences and ensure visual consistency and color balance across channels.
>
> This restructuring ensures that the methodology follows a progressive flow from problem motivation to implementation details, making it easier for readers to understand.
>
> >Q2. Loss Function for Model Training
>
> **A2:** Thank you for your valuable feedback. Our framework optimizes light adaptation using the following objective function:
> The overall objective function is formulated as follows:
>
>  $L_{\mathrm{total}} = \alpha L_{Re} + \beta L_{SSIM} + \gamma L_{HF} + \eta L_{p}$,
>
> where:
> * $L_{Re}$: reconstruction loss,
> * $L_{p}$: perceptual loss,
> *  $L_{HF}$: high-frequency loss,
> * $L_{SSIM}$: structural similarity loss.
>
> $\alpha$, $\beta$, $\gamma$, and $\eta$ are the corresponding weight coefficients.
>
> To justify our design, we conducted an ablation study (Table R2):
>
> Table R2: Ablation studies on different loss functions.
> | $L_{Re}$ | $L_{HF}$ | $L_{SSIM}$ | $L_{p}$ | PSNR↑ | SSIM↑ |
> |:-:|:-:|:-:|:-:|:-:|:-:|
> | &#x2714; | &#x2716; | &#x2714; | &#x2714; | 30.14 | 0.944 |
> | &#x2714; | &#x2714; | &#x2716; | &#x2714; |29.88 | 0.941 |
> | &#x2714; | &#x2714; | &#x2714; | &#x2716;  | 29.72 | 0.939 |
> | &#x2714; | &#x2714; | &#x2714; | &#x2714; | 30.36 | 0.946 |
>
> These results demonstrate that incorporating all four losses leads to the best overall performance, underscoring the necessity of each component. We will clarify this in the revised manuscript.
>
> >Q3. LALNet-Tiny Design Description
>
> **A3:** LALNet-Tiny is a lightweight variant of LALNet, designed to improve efficiency while preserving strong performance. The key differences include:
> * Reduced channel dimensions: Fewer feature channels to reduce computational complexity.
> * Fewer LSSM modules: A more compact design with fewer LSSM modules.
>
> We will include a brief section describing LALNet-Tiny design and relationship to LALNet.
>
> We appreciate the reviewers' valuable insights. Our key improvements include:
> 1. Method Section Restructuring: Introducing explicit subsections and improving explanations with figures.
> 2. Loss Function Clarification: Providing a detailed breakdown, justification, and coefficient selection strategy.
> 3. LALNet-Tiny Performance Analysis: Add a detailed description.
>
> We believe these enhancements will further strengthen our work and address the reviewers' concerns effectively. Thank you for your constructive feedback!

---

> > ### Comment · Reviewer_kmer · 2025-04-03
> >
> > Thanks for the response. Most of my concerns are resolved well.
> >
> > For Q3, could you describe the details of LALNet-tiny, such as exactly how many channels or LSSM modules are reduced?

---

> > > ### Author Response · Authors · 2025-04-03
> > >
> > > We sincerely thank the reviewer for their continued interest in our work. Regarding the details of LALNet-Tiny, we provide the following clarifications:
> > >
> > > In the configuration file, we have two hyperparameters to control the model size:
> > > * **n_feat:** denotes the number of channels;
> > > * **num_blocks:** is a list where the list length denotes the number of stages and num_blocks[i] denotes the number of LSSMs in the i-th stage.
> > >
> > > In the full version of LALNet, the number of channels (n_feat) is 48 and num_blocks is configured as [6, 6, 6, 6]. Specifically, there are totally 4 stages, and each contains 6 LSSM modules.
> > >
> > > In the LALNet-Tiny version, the number of channels (n_feat) is 32 and num_blocks is configured as [1, 1]. That is,  LALNet-Tiny has only 2 stages, each containing one LSSM module.
> > >
> > > These changes allow LALNet-Tiny to maintain good performance while being more computationally efficient. We will include a detailed description of LALNet-Tiny in the revised version. Also, we will release the full source code and pretrained models after the acceptance of this work for further validation and reproducibility.
> > >
> > > If you have any further questions or need more detailed information, please feel free to let us know.

---

### Official Review · Reviewer_2AwK · 2025-03-12

**Overall Recommendation:** 3

**Summary:**

This paper proposes a unified framework, LALNet, for handling multiple light-related tasks in computer vision, such as exposure correction, image retouching, low-light enhancement, and tone mapping. The authors identify common properties across these tasks, such as varying light properties in different color channels and their differences in both spatial and frequency domains. LALNet leverages these properties by using a dual-branch architecture that separates and mixes color channels, incorporating a Light Guided Attention mechanism to ensure visual consistency across channels. The framework is designed to adapt to light variations while preserving image details. Extensive experiments demonstrate that LALNet outperforms state-of-the-art methods in several tasks and requires fewer computational resources.

**Claims And Evidence:**

The motivation of the paper is unclear. In the abstract, the author states that current research addresses light-related tasks, such as HDR imaging and exposure correction, individually. However, it's not clear why handling these tasks separately is a significant drawback. The use of "however" seems to be a transition to introduce the author's work, but logically, there is no strong connection between the two statements, which weakens the paper's fundamental motivation. The writing in other parts of this paper also has not explained this point clearly yet.

**Essential References Not Discussed:**

No essential references are not discussed.

**Experimental Designs Or Analyses:**

Just from the text expounded in this article, the experimental designs and analysis are reasonable.

**Methods And Evaluation Criteria:**

This paper proposes multiple components, such as MCM, DDCM, LGA, and LSSM. However, from Table 5, it seems that LSSM is relatively crucial for performance improvement. Here, LSSM appears to be a structure borrowed from mamba. The author should clearly elaborate and explain this and provide proper citations. In addition, it is not clear what the relationship is between these components and the motivation claimed by the author in Figure 2. The author only lists the necessary materials and steps like writing a recipe or instruction manual (even so, there is still much room for improvement in writing).

**Other Comments Or Suggestions:**

see above.

**Other Strengths And Weaknesses:**

Although the network proposed by the author has achieved a great improvement in final performance, I still have the following concerns.

1. Weak Motivation and Unclear Justification: The manuscript claims that current methods address light-related tasks in isolation, but fails to explain why this is a significant issue. The authors introduce their work by stating "however," but there is no clear link between the shortcomings of existing methods and the necessity of their approach. This weakens the foundational argument for the proposed framework.

2. Modular Approach Lacking Coherent Integration: The paper presents several modules, such as Dual Domain Channel Modulation and Light Guided Attention, but they feel disconnected. While each module may perform well individually, there is insufficient explanation of how these modules are effectively integrated into a unified framework that addresses multiple light-related tasks. This lack of integration diminishes the overall impact of the framework.

3. Writing Style Needs Improvement: The writing style of the manuscript has significant room for improvement. The current structure and presentation make it difficult to grasp the key points clearly. The arguments are often disjointed, and the flow of ideas lacks coherence, which hinders the reader's understanding of the main contributions of the paper.

4. Lack of True Unified Framework: Although the paper claims to present a unified framework, it seems that, if I am not mistaken, the framework still requires separate training for different tasks. In practice, this means that multiple sets of parameters are needed for various tasks, which raises the question of how this approach offers any advantages over training task-specific networks separately. From a practical application standpoint, this limitation undermines the claim of achieving a truly unified solution.

**Questions For Authors:**

see above.

**Relation To Broader Scientific Literature:**

Previous work in this paper has used the unified framework to handle lightness adaptation. This paper should emphasize the differences from these works and further prove the rationality of its own motivation. The current manuscript appears to be a collection of various designed modules without a clear, engaging integration.

**Theoretical Claims:**

There are no proofs for theoretical claims.

---

> ### Author Rebuttal · Authors · 2025-03-30
>
> We sincerely thank the reviewers for their feedback and the opportunity to clarify our work. Below, we address the concerns raised.
> > Q1. Motivation and Justification
>
> **A1:**
> As detailed in the “Introduction” section, many light-related tasks—such as exposure correction, image retouching, low-light enhancement, and tone mapping—share a common goal: adjusting scene lighting to achieve perceptual optimality. However, each task has its own subtle emphasis:
> - **Exposure correction** balances under/overexposed regions.
> - **Image retouching** enhances aesthetics via global illumination adjustments.
> - **Low-light enhancement** brightens dark areas while suppressing noise.
> - **Tone mapping** compresses HDR content while preserving details.
>
> Existing methods typically rely on task-specific architectures and training strategies. We are not claiming that handling these tasks separately is inherently a drawback. Instead, our key motivation is that treating these tasks under a unified framework offers significant advantages:
> 1) **Improved generalization** across diverse lighting conditions (Figure 1).
> 2) **Enhanced efficiency** by reducing the need for multiple specialized models.
> 3) **Consistent performance** across tasks without significant trade-offs (Table 1, Table 2, Table 3, and Table 4).
>
> Indeed, prior works have attempted unified treatments for some of these tasks but often sacrificed performance compared to specialized models. Along this avenue, we aim to design a unified framework that effectively generalizes across diverse light adaptation tasks. Our proposed LALNet achieves this goal by delivering robust performance across multiple light-related tasks (Figure 1).
>
> > Q2. Modular Approach Lacking Coherent Integration
>
> **A2:** LALNet’s design is grounded in two key insights from light-related tasks:
>
> (i) Different color channels have different light properties;
>
> (ii) The channel differences reflected in the spatial and frequency domains are different.
>
> To effectively leverage these observations, LALNet employs a dual-branch architecture:
> * **MCM (Mixed Channel Modulation):** Capture channel-mixed features, focusing on inter-channel relationships and lighting patterns.
> * **DDCM (Dual Domain Channel Modulation):** Extract color-separated features, focusing on light differences and color-specific luminance distributions for each channel in the spatial and frequency domains.
>
> To ensure harmonious fusion of these complementary features, we introduce:
> * **LSSM (Lighting State Space Model):** Integrates color-separated and color-mixed features to enhance illumination consistency across channels.
> * **LGA (Lighting-Guided Attention):** Uses color-separated features as queries, guiding the learning of inter-channel illumination and noise differences for adaptive light restoration.
>
> Our ablation studies (Table 5) confirm that each component is indispensable, as removing any one leads to performance degradation. We will further clarify these interdependencies in the revised manuscript.
>
> > Q4. Unified Framework
>
> **A4:** We appreciate the reviewer’s concern regarding whether LALNet constitutes a truly unified framework. While it is true that each task is optimized separately within LALNet, these tasks share a common architecture and design principles. This shared structure provides a promising advantage over entirely separate task-specific models, considering practical streamlined deployment for low-level tasks. A single framework can be adapted to multiple tasks with minimal adjustments, simplifying hardware deployment. We will clarify this point further in the revised manuscript.
>
> >Q4. Writing and Presentation Improvements
>
> **A4:** We sincerely appreciate the reviewer's feedback regarding the writing style and structure of our manuscript. We acknowledge that certain aspects, such as missing citation ("Mamba"),  the improper use of transitions (e.g., "however"), and disjointed arguments, may have affected the clarity and coherence of our key points. To address these concerns, we will carefully revise the manuscript to enhance the logical flow, strengthen the motivation, and improve the articulation of our contributions and methodology, ensuring they are more accessible to readers.

---

> > ### Comment · Reviewer_2AwK · 2025-04-04
> >
> > Thank you to the authors for the rebuttal, which has addressed some of my concerns.
> >
> > However, in the spirit of constructive dialogue and with a sense of responsibility to the community, I would like to further discuss the motivation of the paper. The authors claim that the proposed unified framework offers three significant advantages.
> >
> > Regarding the first claimed advantage—*improved generation*—Figure 1 shows that the proposed method achieves strong performance on individual tasks. However, it remains unclear how these improvements are related to the claimed *generalization*. As shown in Figure 1, the results are still obtained by training separately on each task. I would encourage the authors to further clarify how they define and evaluate generalization in this context.
> >
> > As for the second point—*enhanced efficiency by reducing the need for multiple specialized models*—this appears to be a claim rather than a demonstrated result. It would strengthen the paper if the authors could provide empirical evidence to substantiate this point.
> >
> > Finally, with respect to the third claimed advantage—consistent performance across tasks without significant trade-offs (Table 1, Table 2, Table 3, and Table 4)—I would appreciate further clarification on whether this consistency is a unique property of the proposed approach. Alternatively, do prior methods also maintain such consistency without exhibiting clear trade-offs in similar settings?
> >
> > Overall, based on the author's response, the paper appears to present a modest contribution—one with some value, but not particularly novel. It seems to offer a relatively incremental and conventional network improvement, which may be seen as somewhat routine within the current landscape.

---

> > > ### Author Response · Authors · 2025-04-04
> > >
> > > We sincerely appreciate the reviewer's constructive feedback and the opportunity to clarify our contributions further. Below, we address each point in detail:
> > > ### 1. Clarification on Generalization
> > > In our manuscript, we introduce LALNet as a **unified framework**. But, through the discussions with you and other reviewers, we have come to recognize that this term may not fully convey the unique characteristics of our approach. To provide a clearer understanding, we will revise the terminology in the manuscript and describe LALNet as a **multi-task framework**. It is important to clarify that LALNet is not an all-in-one model, i.e., one model performs well in multiple tasks. Instead, LALNet operates as a multi-task framework, where each task is trained independently while leveraging the same underlying architecture. This design allows the models to take advantage of the shared architecture, thereby simplifying hardware deployment in multitasking scenarios.
> > >
> > > Additionally, as shown in Table R4, we evaluate LALNet on third-party datasets. The results demonstrate that LALNet exhibits promising adaptability to unseen data, further validating its effectiveness as a multi-task framework and its consistent performance across diverse tasks.
> > >
> > > Table R4: Validating generalization on third-party datasets includes HDR Survey and UVTM video datasets.
> > > |Datasets|Metrics|HDRNet|CLUT|CoTF|ZS-Diffusion|IVTMNet|Ours|
> > > |:-:|:-:|:-:|:-:|:-:|:-:|:-:|:-:|
> > > |HDR Survey|TMQI|0.8641|0.8140|0.8612|0.8915|0.9160|0.9296|
> > > |UVTM|TMQI|0.8281|0.8799|0.9006|0.8871|0.8991|0.9584|
> > > ### 2. Empirical Evidence for Efficiency Improvements
> > > **From the deployment perspective:** LALNet requires the design of only one hardware path, loading different models for different tasks. In contrast, multiple task-specific models require distinct hardware paths, each with different operators and resource allocations, which complicates deployment, especially on edge devices. This simplicity is especially beneficial in resource-constrained environments, where efficient deployment is critical.
> > >
> > > **Performance as a single model:** As shown in Table R5, the comparison between multiple task-specific models and LALNet demonstrates that LALNet significantly reduces computational cost in terms of FLOPs, parameters, and inference time.
> > >
> > > Table R5: Computational cost comparison.
> > > |Method|#Params|GFLOPs|Run Time|
> > > |:-:|:-:|:-:|:-:|
> > > |RetinexMamba|4.59M|42.8|0.478|
> > > |RetinexFormer|1.61M|15.6|0.028|
> > > |MambaIR|4.31M|110.1|0.460|
> > > |Ours| 230K|1.8|0.018|
> > > ### 3. Clarification on the Consistency
> > > As shown in Fig. 1 and Tab. 1–4, LALNet achieves significant performance gains across all four tasks:
> > > * Exposure correction: +0.86dB
> > > * Image retouching: +2.15dB
> > > * Low-light enhancement: +1.23dB
> > > * Tone mapping: +3.86dB
> > >
> > > These results demonstrate that LALNet achieves consistent performance across all tasks, unlike some existing methods, such as in the exposure correction task, Retinexformer 23.75 vs. 23.56 RetinexMamba, but in the image retouching task, Retinexformer 26.20 vs. 26.81 RetinexMamba. And COTF (CVPR 2024), which performs well (ranked $3^{rd}$ in Table 1) in exposure correction but shows a drop in performance for image retouching (ranked $10^{th}$ in Table 2) and low-light (ranked $10^{th}$ in Table 4) tasks.
> > >
> > > The consistency between tasks in LALNet is due to our careful observation of these tasks, and based on the physical properties of the light-related tasks, we designed different modules to ensure the light consistency between different tasks.
> > > ### 4. Clarification of Innovation
> > > Compared to existing methods, LALNet introduces significant innovations rooted in a careful analysis of light-related tasks. Based on these insights, we developed a multi-task framework specifically tailored for handling multiple light-related tasks.
> > >
> > > Our framework is grounded in the physical properties of light-related tasks, where different color channels exhibit unique lighting characteristics. We design the DDCM module to extract color-separated features, focusing on light differences and color-specific luminance distributions in both spatial and frequency domains. Additionally, the LSSM integrates both color-separated and color-mixed features, enhancing illumination consistency across channels. By using the extracted color-separated features as queries, LGA guides the main channel-mixed features to learn channel differences, improving adaptability for task-specific needs.
> > >
> > > These physics-based design innovations are first introduced in our work and have been validated across multiple tasks to demonstrate their effectiveness. We believe that LALNet represents a meaningful step forward, offering a novel approach to task modeling grounded in the physical properties of light, rather than merely an incremental improvement.
> > >
> > > We appreciate the reviewer’s thoughtful comments and will incorporate these clarifications. If you have any further questions or need more detailed information, please feel free to let us know.

---

### Decision · Program_Chairs · 2025-05-01

**Decision:**

Accept (poster)

**Comment:**

This paper presents LALNet, a unified framework designed to tackle multiple light-related image processing tasks such as exposure correction, image retouching, low-light enhancement, and tone mapping. Through extensive experiments, the authors demonstrate that LALNet outperforms state-of-the-art methods across several benchmarks while maintaining computational efficiency.

The idea of unifying these tasks into a single model is practically valuable, and reviewers acknowledged the empirical performance shown in experimental results.


Reviewers initially raised concerns regarding the motivation for the architectural design choices, the clarity of the representation strategy, and the depth of the ablation analysis. However, the authors provided a detailed and convincing rebuttal, addressing these issues with additional explanation and justification.

That said, the level of innovation remains moderate, as many of the modules and design elements are adapted from existing works. While the unified framework is well-engineered and well-validated, the contribution may not be sufficiently novel particularly for the ICML audience, given the fact that there is no theorical contribution.